# Synthesis and Antimicrobial Studies of Coumarin-Substituted Pyrazole Derivatives as Potent Anti-*Staphylococcus aureus* Agents

**DOI:** 10.3390/molecules25122758

**Published:** 2020-06-15

**Authors:** Rawan Alnufaie, Hansa Raj KC, Nickolas Alsup, Jedidiah Whitt, Steven Andrew Chambers, David Gilmore, Mohammad A. Alam

**Affiliations:** 1Department of Chemistry and Physics, College of Science and Mathematics, Arkansas State University, Jonesboro, AR 72467, USA; rawan.alnufaie@smail.astate.edu (R.A.); hansa.kc@smail.astate.edu (H.R.K.C.); nickolas.alsup@smail.astate.edu (N.A); jedidiah.whitt@smail.astate.edu (J.W.); steven.chambers@smail.astate.edu (S.A.C.); 2Department of Biological Sciences, College of Science and Mathematics, Arkansas State University, Jonesboro, AR 72467, USA; dgilmore@astate.edu

**Keywords:** pyrazole, coumarin, antimicrobial, hydrazone, biofilm, *Staphylococcus aureus*, *S. epidermidis*, MRSA

## Abstract

In this paper, synthesis and antimicrobial studies of 31 novel coumarin-substituted pyrazole derivatives are reported. Some of these compounds have shown potent activity against methicillin-resistant *Staphylococcus aureus* (MRSA) with minimum inhibitory concentration (MIC) as low as 3.125 µg/mL. These molecules are equally potent at inhibiting the development of MRSA biofilm and the destruction of preformed biofilm. These results are very significant as MRSA strains have emerged as one of the most menacing pathogens of humans and this bacterium is bypassing HIV in terms of fatality rate.

## 1. Introduction

Drug-resistant infections kill more than 700,000 people each year and the number could increase to 10 million per year by 2050 in the world. The menace of antimicrobial resistance could force up to 24 million people into extreme poverty [1]. According to a Centers for Disease Control and Prevention (CDC) report, each year more than 2.8 million antibiotic infections occur and more than 35,000 people die as a result of these infections in the United States alone. One of the four guidelines recommended by CDC to combat antibiotic resistance is promoting the development of new antibiotics and new diagnostic tests for dealing with drug-resistant bacteria [2]. *Staphylococcus aureus* and its drug-resistant variants are among the most common bacterial pathogens. Methicillin-resistant *S. aureus* (MRSA) causes ten-fold more infections than all multidrug resistant Gram-negative bacteria. A special pathogenic feature of *S aureus* is its ability to survive on abiotic and biotic surfaces in a slimy extracellular matrix called biofilm. The bacteria comprising biofilms are more resistant to antibiotics and the host immune system by a variety of mechanisms [3].

Pyrazole, a five-membered 1,2-diazole, is found in a number of widely used drugs such as celecoxib and lonazolac [4]. The pyrazole nucleus is less susceptible to oxidative metabolism than other five-membered heterocycles [5]. Derivatives of this diazole have been reported as antimicrobial agents in a number of publications by us [6,7,8,9,10,11] and others [12,13]. Similarly, natural and synthetic coumarin derivatives have attracted considerable interest because of their various pharmacological activities (Figure 1) [4]. Warfarin and novobiocin are examples of widely used coumarin-derived medications. Coumarin derivatives have been reported as potent growth inhibitors of a wide variety of bacteria such as methicillin-resistant *S. aureus* (MRSA) [14], *Pseudomonas aeruginosa* [15], *Escherichia coli* [16], and several other bacterial species [17].

In our efforts to develop potent antimicrobial agents, we have reported the synthesis and antimicrobial studies of phenyl [6,10], fluorophenyl [8,9], and naphthyl-substituted pyrazole derivatives [11]. We recently reported the synthesis, antimicrobial, and toxicological studies of coumarin-substituted pyrazole-derived hydrazones. Some of the molecules are potent growth inhibitors of *Acinetobacter baumannii* with an MIC value as low as 1.56 µg/mL. These potent molecules are non-toxic to human healthy and cancer cell lines and in vivo mouse model studies [7]. Encouraged with these results, we synthesized and studied the antimicrobial properties of 31 new fluoro and hydroxy-substituted coumarin-derived pyrazole compounds. These new coumarin-pyrazole-hydrazone derivatives are readily synthesized by using commercially available starting materials and reagents using benign reaction conditions.

## 2. Results and Discussion

### 2.1. Chemistry

The reaction of 4-hydrazinobenzoic acid (**1**) with fluoro (**2a**) and hydroxy (**2b**) substituted 3-acetylcoumarin formed the corresponding hydrazones (**3a–b**), which were subjected to further reaction with POCl_3_/DMF to get the formyl-substituted pyrazole derivatives (**4a–b**) in overall excellent yields (Scheme 1). Multi-gram scale synthesis of pure pyrazole-derived aldehydes (**4a–b**) has helped to synthesize a series of hydrazone derivatives (Scheme 2).

The reaction of phenylhydrazine with aldehyde derivatives (**4a**) formed the product (**5**) in good yield (Table 1). *N,N*-Disubstituted hydrazine derivative also reacted with the aldehyde derivative (**4**) to give the desired compounds (**6**, **7**, and **8**) in 77%, 84%, and 82%, yields, respectively (Table 1). Products containing electron-donating groups (**9** and **10**) were obtained in 89% and 79% yields, respectively. Moderately electron-withdrawing groups: fluoro (**11**), chloro (**12**), and bromo (**13**) substituted coumarin-derived hydrazone products were formed in very good yield. Dihalogen substituted compounds formed accordingly and the pure products (**14**, **15**, **16**, **17**, and **18**) were isolated in 75–95% yield. Strong electron-withdrawing groups such as trifluoromethyl (**19**), cyano (**20**), and nitro (**21**) formed the corresponding hydrazones. Reaction of *N,N*-dimethylhydrazine also formed the product (**22**) efficiently (Table 1).

Similarly, hydroxy-substituted coumarin derivative (**4b**) reacted with the hydrazines smoothly to form the corresponding hydrazones (**23**–**35**) efficiently (Table 2). The *N*-phenyl-derived product (**23**) formed in 79% yield. 3-Fluoro (**24**), 4-chloro (**25**), and 4-bromo (**26**) derivatives formed in 86%, 71%, and 80% yields, respectively. Difluoro (**27** and **28**) and dichloro (**29**) hydrazones were synthesized in 75% average yield. Mixed halo compounds (**30** and **31**) were formed efficiently. Strong electron-withdrawing groups such as cyano (**32**) and nitro (**33**) also formed in very good yield. Last but not least, *N*-carbamate product (**34**) formed in 72% yield.

### 2.2. Antimicrobial Studies

Synthesized molecules were tested against eight strains of Gram-positive bacteria and three strains of *A. baumannii*, a Gram-negative bacterium (Table 3). *N*-Phenyl and *N*-methyl-*N*-phenyl substituted compounds (**5** and **6**) did not show any significant activity up to 50 µg/mL against the tested bacterial strains. Excitingly, *N*,*N*-diphenyl substituted compound (**7**) showed potent activity against all the tested Gram-positive bacterial strains but no activity against *A. baumannii*. This molecule (**7**) inhibited the growth of methicillin-sensitive *S. aureus* (MSSA) with an MIC value as low as 3.125 µg/mL. Four methicillin-resistant strains were inhibited by this molecule with an MIC value as low as 1.56 µg/mL. It is worth mentioning that three of the MRSA strains: *S. aureus* ATCC 33591 (Sa91), *S. aureus* ATCC 700699 (Sa99), and *S. aureus* ATCC 33592 (Sa92) possess the SCC mec II or III genomic islands conferring multidrug resistance. This molecule also inhibited the growth of *S. epidermidis* 700296 (Se), and *B. subtilis* ATCC 6623 (Bs) with an MIC value of 6.25 µg/mL. Replacing one phenyl group with a benzyl group eliminated the activity of the resultant molecule (**8**). Methyl (**9**) and ethyl (**10**) substitution showed partial activity against the tested strains. Fluoro-phenyl substituted compound (**11**) did not show any growth inhibition of the tested bacteria. Chloro (**12**) and bromo (**13**) substitution showed moderate activity against Gram-positive bacterial strains. These two molecules (**12** and **13**) also showed similar activity against three *A. baumannii* strains with MIC value as low as 6.25 µg/mL. 2,5-Difluoro substitution (**14**) failed to show any activity against the tested bacteria. Nevertheless, 3,4-difluoro substitution (**15**) showed moderate activity against both the Gram-positive and Gram-negative strains with MIC value of 6.25 µg/mL. Other dihalogen-substituted compounds (**16**, **17**, and **18**) did not show any antimicrobial potency. Very strong electron withdrawing groups such as trifluoromethyl (**19**) showed good activity against Gram-positive strains with MIC value as low as 3.125 µg/mL although the other two compounds (**20** and **21**) with very strong electron withdrawing substitution did not show any activity. *N,N*-Dimethyl compound (**22**) also failed to show any activity against the tested strains. Surprisingly, hydroxy-substituted coumarin compounds (**23**–**34**) did not show any activity

Based on these results, we can deduce the following structure activity relationship. Fluoro-substitution in the coumarin moiety produced potent antimicrobial compounds, whereas the hydroxy-substitution almost eliminated the activity of the molecules. This finding could be due to the different nature of the hydroxy and fluoro substituents. Among the fluoro-substituted coumarins, *N,N*-diphenyl derivative showed potent activity but replacing one of the phenyl groups with methyl (**6**) or benzyl (**8**) eliminated the potency of the compounds. This observation is in agreement with our previous report [7]. Hydrophobic mono-substituted compounds showed better activity than the disubstituted compounds, 4-trifluoromethyl substituent (**19**) showed the most potent overall activity among all the compounds against the tested Gram-positive bacteria. Substituent with a very strong electron withdrawing nature such as cyano (**20**) and nitro (**21**) eliminated activity of the compounds.

After finding potent molecules against planktonic bacteria, we studied their ability to affect *S. aureus* biofilms (Figure 2). The most potent compound (**7**) inhibited biofilm by 85% at 0.5 × MIC. Higher concentrations were even more effective. Similarly, compound **12** inhibited the biofilm formation at 2× and 1 × MIC values but its potency decreased at 0.5 × MIC. Compounds (**13** and **19**) also inhibited biofilm formation similarly to compound (**7**). The positive control, vancomycin, also inhibited biofilm formation at 2× and 1×MICs but showed significantly less effectiveness at 0.5 × MIC. It should be noted that inhibition of bacterial growth would be expected to result in reduced biofilm formation. Therefore, it is very encouraging that significant inhibition of biofilm formation by our compounds occurred at a concentration of 0.5 × MIC. Three of these potent compounds (**7**, **12**, and **13**) were efficient eliminators of preformed biofilms of *S. aureus*. Compound **19** removed more than 90% of the biofilm at 2× and 1 × MICs, although 0.5 × MIC showed some reduced effectiveness against the preformed biofilm of *S. aureus*. These results are very significant as the positive control could only eliminate ~70% preformed biofilm at 2 × MIC.

The two most effective compounds against *A. baumannii* (**12**, **13**) and the two most effective against *S. aureus* (**7**, **19**) were used in time kill assays over a 24-h period. At 4 × MIC, both (**12**) and (**13**) were bacteriostatic against *A. baumannii*. Compound (**19**) was likewise bacteriostatic against the MRSA strain tested, but compound (**7**) had marked bactericidal activity, reducing colony-forming units (CFU)/mL to undetectable by 4 h as shown in Figure 3.

We tested the potent compounds for their possible toxicity for the human embryonic kidney cell line (HEK293). Potent compounds (**7**, **12**, and **13**) inhibited the growth of cell lines with an IC_50_ around 15 µg/mL but compound **19** showed much less toxicity. More than 90% of cells survived at 25 µg/mL (Figure 4).

## 3. Materials and Methods

All of the reactions were carried out under an air atmosphere in round-bottom flasks. Reagents, substrate, and solvents for reactions, recrystallizations, and deuterated solvents for ^1^H and ^13^C-NMR spectroscopy were purchased from Fisher Scientific (Hanover Park, IL, USA) and Oakwood chemical (Estill, SC, USA). The more details of spectroscopy in Appendix A.

A mixture of 4-hydrazinobenzoic acid (**1**, 10 mmol, 1.521 g) and 3-acetylcoumarin derivatives (**2a–b**, 10.5 mmol) in ethanol was refluxed for 8 h to obtain the hydrazone derivative (**3**) (Scheme 1). The solvent was evaporated under the reduced pressure at 60 °C, and the hydrazone derivative was further dried in vacuo. The dried product was used for further reactions without isolation or purification. The hydrazone derivative (**3**) was dissolved in *N,N*-dimethyl formamide (DMF, 30 mL) and the flask was sealed by a rubber septum. The mixture was stirred for 15 min to dissolve the solid material completely. The clear solution was stirred at 0 °C in an ice bath, and phosphorous oxychloride (POCl_3_, 5.43 mL) was added dropwise to form the Vilsmeier reagent. After 30 min, the reaction mixture was heated for 8 h at 90 °C. After the completion of the reaction, the mixture was poured onto ice in a beaker and then stirred for 12 h to precipitate the product, which was filtered and washed with water repeatedly until the filtrate was clear. The final product was dried under vacuum.

Novel coumarin-derived hydrazones were synthesized by the reaction of the aldehyde product (**3a** and **3b**, 1 mmol) with commercially available substituted-hydrazines (1.1 mmol) in ethanol (Scheme 2). Sodium acetate (1.1 mmol, 0.088 g) and acetic acid were added in case of the hydrochloride salt of the hydrazine derivatives. The resulting product was filtered and washed with ethanol (~15 mL) followed by washing with water (~20 mL) to get the pure product.

### 3.1. Experimental Data

*4-[3-(7-fluoro-2-oxo-3,8a-dihydrochromen-3-yl)-4-formyl-pyrazol-1-yl]benzoic acid* (**4**). Brownish; (3.567 g, 93.7%) ^1^H-NMR (300 MHz, DMSO-*d*_6_): δ 9.91 (s, 1H), 9.35 (s, 1H), 8.68 (s, 1H), 8.09 (s, 4H), 7.72 (d, *J* = 8.0 Hz, 1H), 7.55-7.53 (m, 2H); ^13^C-NMR (75 MHz, DMSO-*d*_6_): δ 186.0, 166.9, 158.6 (d, *^1^J* = 239.6 Hz), 159.5, 150.3, 148.0, 142.6, 141.8, 133.6, 131.4, 130.1, 124,2, 121.1, 120.4 (d, *^2^J* = 24.5 Hz), 120.1 (d, *^3^J* = 9.7 Hz), 119.4, 118.7 (d, *^3^J* = 8.7 Hz), 114.6 (d, *^2^J* = 24.3 Hz). HRMS (ESI-FTMS) Mass (*m*/*z*): calcd for C_20_H_11_FN_2_O_5_ [M + H]^+^ = 379.0725, found 379.0738.

*4-[3-(7-fluoro-2-oxo-3,8a-dihydrochromen-3-yl)-4-[(E)-(phenylhydrazono)methyl]pyrazol-1-yl]benzoic acid* (**5**). Yellow; (343 mg, 73%) ^1^H-NMR (300 MHz, DMSO-*d*_6_): δ 10.22 (s, 1H), 8.98 (s, 1H), 8.32 (s, 1H), 8.12–8.05 (m, 4H), 7.84 (s, 1H), 7.74 (d, *J* = 9.3 Hz, 1H), 7.60–7.58 (m, Hz, 2H), 6.99 (t, *J* = 8.1 Hz, 2H), 6.75–6.73 (m, 2H), 6.63 (t, *J* = 7.2 Hz, 1H); ^13^C-NMR (75 MHz, DMSO-*d*_6_): δ 167.0, 158.6 (d, *^1^J* = 239.3 Hz), 159.1, 150.2, 146.0, 145.6, 142.4, 141.9, 131.4, 129.2, 128.99, 128.96, 127.3, 123.3, 121.3, 120.3 (d, *^3^J* = 9.5 Hz), 120.0 (d, *^2^J* = 24.6 Hz), 118.8, 118.5 (d, *^3^J* = 8.5 Hz), 118.3, 114.4 (d, *^2^J* = 24.2 Hz), 112.0. HRMS (ESI-FTMS) Mass (*m*/*z*): calcd for C_26_H_20_N_4_O_5_ [M + H]^+^ = 469.1506, found 469.1298.

*4-[3-(6-fluoro-2-oxo-3,8a-dihydrochromen-3-yl)-4-[(E)-[methyl(phenyl)hydrazono]methyl]pyrazol-1-yl]benzoic acid* (**6**). Yellow; (373 mg, 77%) ^1^H-NMR (300 MHz, DMSO-*d*_6_): 8.95 (s, 1H), 8.32 (s, 1H), 8.09 (s, 4H), 7.76–7.72 (m, 1H), 7.62–7.56 (m, 3H), 7.10–7.07 (m, 2H), 6.98 (t, *J* = 7.3 Hz, 2H), 6.75 (t, *J* = 7.2 Hz, 1H), 2.50–2.49 (m, 3H); ^13^C-NMR (75 MHz, DMSO-*d*_6_): δ 167.1, 158.6 (d, *^1^J* = 239.3 Hz), 159.2, 150.3, 147.5, 146.0, 142.4, 141.8, 131.4, 129.0, 128.9, 127.2, 125.4, 123.7, 122.2, 120.3 (d, *^3^J* = 9.5 Hz), 120.1, 119.7, 118.6 (d, *^3^J* = 8.5 Hz), 118.3, 114.5, 114.2, 32.7. HRMS (ESI-FTMS) Mass (*m*/*z*): calcd for C_27_H_19_FN_4_O_4_ [M + H]^+^ = 483.1463, found 483.1466.

*4-[4-[(E)-(diphenylhydrazono)methyl]-3-(6-fluoro-2-oxo-3,8a-dihydrochromen-3-yl)pyrazol-1-yl]benzoic acid* (**7**). Yellow; (459 mg, 84%) ^1^H-NMR (300 MHz, DMSO-*d*_6_): δ 8.94 (s, 1H), 8.33 (s, 1H), 8.09–8.06 (m, 2H), 8.01–7.98 (m, 1H), 7.77–7.74 (m, 1H), 7.65–7.58 (m, 2H), 7.30–7.24 (m, 4H), 7.16–7.11 (m, 3H), 6.94–6.91 (m, 4H); ^13^C-NMR (75 MHz, DMSO-*d*_6_): δ 167.0, 158.6 (d, *^1^J* = 239.3 Hz), 159.0, 150.2, 145.8, 143.1, 142.3, 141.9, 131.4, 130.2, 129.0, 128.6, 128.0, 124.9, 123.7, 122.2, 121.0, 120.2 (d, *^3^J* = 9.7 Hz), 119.8, 118.6 (d, *^3^J* = 8.7 Hz), 118.3, 144.4 (d, *^2^J* = 24.3 Hz). HRMS (ESI-FTMS) Mass (*m*/*z*): calcd for C_32_H_21_FN_4_O_4_ [M + H]^+^ = 545.1620, found 545.1613.

*4-[4-[(E)-[benzyl(phenyl)hydrazono]methyl]-3-(6-fluoro-2-oxo-3,8a-dihydrochromen-3-yl)pyrazol-1-yl]benzoic acid* (**8**). Yellow; (459 mg, 82%) ^1^H-NMR (300 MHz, DMSO-*d*_6_): δ 8.88 (s, 1H), 8.28 (s, 1H), 8.09–8.00 (m, 4H), 7.73–7.71 (m, 1H), 7.58 (s, 3H), 7.53–7.12 (m, 5H), 6.99–6.86 (m, 4H), 6.75–6.71 (m, 1H), 5.20 (s, 2H); ^13^C-NMR (75 MHz, DMSO-*d*_6_): δ 167.0, 158.6 (d, *^1^J* = 239.4 Hz), 159.0, 150.2, 147.2, 145.9, 142.4, 141.7, 136.2, 131.4, 129.1, 128.9, 127.8, 127.4, 126.6, 125.3, 123.9, 121.8, 120.3, 120.2, 119.9 (d, *^2^J* = 24.6 Hz), 118.6 (d, *^3^J* = 8.7 Hz), 118.3, 114.4 (d, *^2^J* = 24.1 Hz), 113.9. 48.2. HRMS (ESI-FTMS) Mass (*m*/*z*): calcd for C_33_H_23_FN_4_O_4_ [M + H]^+^ = 559.1776, found 559.1768.

*4-[3-(7-fluoro-2-oxo-3,8a-dihydrochromen-3-yl)-4-[(E)-(p-tolylhydrazono)methyl]pyrazol-1-yl]benzoic acid* (**9**). Orange; (481 mg, 89%) ^1^H-NMR (300 MHz, DMSO-*d*_6_): δ 10.10 (s, 1H), 8.96 (s, 1H), 8.31 (s, 1H), 8.11–8.05 (m, 4H), 7.79–7.72 (m, 2H), 7.60–7.59 (m, 2H), 6.83–6.80 (m, 2H), 6.67–6.65 (m, 2H), 2.14 (s, 3H); ^13^C-NMR (75 MHz, DMSO-*d*_6_): δ 167.1, 158.6 (d, *^1^J* = 239.4 Hz), 159.1, 150.3, 145.9, 143.3, 142.4, 141.9, 131.4, 129.7, 128.9, 128.3, 127.3, 127.1, 123.3, 121.4, 120.3 (d, *^3^J* = 9.6 Hz), 120.0 (d, *^2^J* = 24.2 Hz), 118.5 (d, *^3^J* = 8.4 Hz), 118.3, 114.4 (d, *^2^J* = 24.0 Hz), 112.0, 20.6. HRMS (ESI-FTMS) Mass (*m*/*z*): calcd for C_27_H_19_FN_4_O_4_ [M + H]^+^ = 483.1463, found 483.1455.

*4-[4-[(E)-[(2-ethylphenyl)hydrazono]methyl]-3-(7-fluoro-2-oxo-3,8a-dihydrochromen-3-yl)pyrazol-1-yl]benzoic acid* (**10**). Orange; (393 mg, 79%)^1^H-NMR (300 MHz, DMSO-*d*_6_): 9.52 (s, 1H), 9.00 (s, 1H), 8.32 (s, 1H), 8.12–8.10 (m, 5H), 7.75–7,72 (m, 1H), 7.60–7.58 (m, 2H), 6.99–6.91 (m, 2H), 6.70–6.60 (m, 2H), 2.56–2.49 (m, 2H), 1.11 (t, *J* = 7.3 Hz, 3H); ^13^C-NMR (75 MHz, DMSO-*d*_6_): δ 167.1, 158.6 (d, *J* = 239.3 Hz), 159.1, 150.2, 146.1, 142.7, 142.4, 142.0, 131.4, 130.2, 129.9, 128.9, 128.6, 127.3, 126.5, 123.4, 121.5, 120.3 (d, *^3^J* = 9.6 Hz), 119.8, 119.0, 118.6 (d, *^3^J* = 8.8 Hz), 118.3, 114.4 (d, *^2^J* = 24.3 Hz), 112.3, 23.5, 14.3. HRMS (ESI-FTMS) Mass (*m*/*z*): calcd for C_28_H_21_FN_4_O_4_ [M + H]^+^ = 497.1620, found 497.1609.

*4-[3-(7-fluoro-2-oxo-3,8a-dihydrochromen-3-yl)-4-[(E)-[(3-fluorophenyl)hydrazono]methyl]pyrazol-1-yl]benzoic acid* (**11**). Orange; (395 mg, 81%) ^1^H-NMR (300 MHz, DMSO-*d*_6_): 10.45 (s, 1H), 9.03 (s, 1H), 8.34 (s, 1H), 8.09–8.06 (m, 4H), 7.86 (s,1H), 7.73 (d, *J* = 7.1 Hz, 1H), 7.58–7.56 (m, 2H), 7.09–7.01 (m, 1H), 6.56–6.51 (m, 2H), 6.39 (t, *J* = 6.3 Hz, 1H); ^13^C-NMR (75 MHz, DMSO-*d*_6_): 167.0, 163.7 (d, *^1^J* = 239.0 Hz), 158.6 (d, *^1^J* = 239.4 Hz), 159.1, 150.2, 147.6 (d, *^3^J* = 11.0 Hz), 146.0, 142.4, 142.1, 131.4, 130.8 (d, *^3^J* = 9.9 Hz), 130.3, 129.0, 127.7, 123.1, 121.0, 120.2 (d, *^3^J* = 9.7 Hz), 119.8, 118.6 (d, *^3^J* = 8.5 Hz), 118.4, 114.4 (d, *^2^J* = 24.1 Hz), 108.2, 104.8 (d, *^2^J* = 21.3 Hz), 98.5 (d, *^2^J* = 26.1 Hz). HRMS (ESI-FTMS) Mass (*m*/*z*): calcd for C_26_H_16_F_2_N_4_O_4_ [M + H]^+^ = 487.1212, found 487.1198.

*4-[4-[(E)-[(3-chlorophenyl)hydrazono]methyl]-3-(7-fluoro-2-oxo-3,8a-dihydrochromen-3-yl)pyrazol-1-yl]benzoic acid* (**12**). Orange; (429 mg, 85%) ^1^H-NMR (300 MHz, DMSO-*d*_6_): 10.42 (s, 1H), 9.02 (s, 1H), 8.34 (s, 1H), 8.12–8.05 (m, 4H), 7.88 (s, 1H), 7.74–7.72 (m, 1H), 7.58–7.56 (m, 2H), 7.05 (t, *J* = 8.0 Hz, 1H), 6.77 (s, 1H), 6.66 (t, *J* = 8.1 Hz, 2H); ^13^C-NMR (75 MHz, DMSO-*d*_6_): δ 167.0, 158.6 (d, *^1^J* = 239.3 Hz), 159.1, 150.3, 147.0, 146.0, 142.4, 142.1, 134.3, 131.4, 130.7 (d, *^3^J* = 22.5 Hz), 129.0, 127.9, 123.1, 120.9, 120.3, 120.1, 119.8, 118.8 (d, *^3^J* = 8.7 Hz), 118.4, 118.2, 114.4 (d, *^2^J* = 24.3 Hz), 111.1, 110.8. HRMS (ESI-FTMS) Mass (*m*/*z*): calcd for C_26_H_16_ClFN_4_O_4_ [M + H]^+^ = 503.0917, 505.0890, found 503.0902, 505.0905.

*4-[4-[(E)-[(3-bromophenyl)hydrazono]methyl]-3-(7-fluoro-2-oxo-3,8a-dihydrochromen-3-yl)pyrazol-1-yl]benzoic acid* (**13**). Orange; (428 mg, 78%) ^1^H-NMR (300 MHz, DMSO-*d*_6_): δ 10.40 (s, 1H), 9.02 (s, 1H), 8.33 (s, 1H), 8.12–8.05 (m, 4H), 7.88 (s, 1H), 7.74–7.71 (m, 1H), 7.58–7.52 (m, 2H), 7.01–7.93 (m, 2H), 6.79-6.70 (m, 2H); ^13^C-NMR (75 MHz, DMSO-*d*_6_): δ 167.0, 158.6 (d, *^1^J* = 239.3 Hz), 159.1, 150.3, 147.1, 146.0, 142.4, 142.1, 131.4, 131.2, 130.6, 129.0, 127.9, 123.1, 122.9, 121.1, 120.8, 120.2 (d, *^3^J* = 9.7 Hz), 119.8, 118.9 (d, *^3^J* = 8.4 Hz), 118.4, 114.4 (d, *^2^J* = 24.1 Hz), 113.9, 111.2. HRMS (ESI-FTMS) Mass (*m*/*z*): calcd for C_26_H_16_BrFN_4_O_4_ [M + H]^+^ = 547.0412, 549.0393, found 547.0400, 549.0394.

*4-[4-[(E)-[(2,5-difluorophenyl)hydrazono]methyl]-3-(6-fluoro-2-oxo-3,8a-dihydrochromen-3-yl)pyrazol-1-yl]benzoic acid* (**14**). Orange; (379 mg, 75%) ^1^H-NMR (300MHz, DMSO-*d*_6_): 10.38 (s, 1H), 9.08 (s, 1H), 8.33 (s, 1H), 8.14–8.05 (m, 5H), 7.72 (d, *J* = 7.2 Hz, 1H), 7.60–7.50 (m, 2H), 7.13–7.05 (m, 1H), 6.75–6.68 (m, 1H), 6.45–6.38 (m, 1H); ^13^C-NMR (75 MHz, DMSO-*d*_6_): δ 167.1, 159.5 (d, *^1^J* = 236.5 Hz), 158.6 (d, *^1^J* = 239.5 Hz), 159.1, 150.2, 145.4 (d, *^1^J* = 233.5 Hz), 146.1, 142.3, 142.2, 135.1 (t, ^3^*J* = 11.8 Hz), 133.1, 131.4, 129.2, 128.2, 122.9, 120.7, 120.2, 120.0 (d, *^3^J* = 13.1 Hz), 118.6 (d, *^3^J* = 8.6 Hz), 118.4, 116.4-116.0 (m), 114.4 (d, *^2^J* = 24.2 Hz), 103.8 (d, *^3^J* = 17.7 Hz), 100.2 (d, *^2^J* = 30.0 Hz). HRMS (ESI-FTMS) Mass (*m*/*z*): calcd for C_26_H_15_F_3_N_4_O_4_ [M + H]^+^ = 505.1118, found 505.1110.

*4-[4-[(E)-[(3,4-difluorophenyl)hydrazono]methyl]-3-(6-fluoro-2-oxo-3,8a-dihydrochromen-3-yl)pyrazol-1-yl]benzoic acid* (**15**). Orange; (425 mg, 84%) ^1^H-NMR (300 MHz, DMSO-*d*_6_): 10.40 (s, 1H), 9.02 (s, 1H), 8.33 (s, 1H), 8.11–8.04 (m, 4H), 7.84 (s, 1H), 7.75–7.72 (m, 1H), 7.57 (d, *J* = 5.8 Hz, 2H), 7.16–7.07 (m, 1H), 6.71–6.65 (m, 1H), 6.52–6.49 (m, 1H); ^13^C-NMR (75 MHz, DMSO-*d*_6_): δ 167.0, 158.6 (d, *^1^J* = 239.4 Hz), 159.2, 150.4 (dd, *J* = 13.0, 240.6 Hz), 150.6, 146.0, 142.8 (dd, *J* = 12.8, 233.6 Hz), 143.2 (d, *^3^J* = 8.4 Hz), 142.4, 142.1, 131.4, 130.4, 129.0, 127.7, 122.9, 120.9, 120.2 (d, *^3^J* = 9.5 Hz), 120.25, 119.9, 118.6 (d, *^3^J* = 8.5 Hz), 118.4, 118.1, 117.9, 114.4 (d, *^2^J* = 24.1 Hz). HRMS (ESI-FTMS) Mass (*m*/*z*): calcd for C_26_H_15_F_3_N_4_O_4_ [M + H]^+^ = 505.1118, found 505.1106.

*4-[4-[(E)-[(2,4-dichlorophenyl)hydrazono]methyl]-3-(6-fluoro-2-oxo-3,8a-dihydrochromen-3-yl)pyrazol-1-yl]benzoic acid* (**16**). Yellowish; (415 mg, 77%)^1^H-NMR (300 MHz, DMSO-*d*_6_): 13.11 (br s, 1H), 9.98 (s, 1H), 9.08 (s, 1H), 8.34 (s, 1H), 8.21 (s, 1H), 8.10 (s, 4H), 7.76–7.72 (m, 1H), 7.62–7.59 (m, 2H), 7.418–7.410 (m, 1H), 7.12–7.09 (s, 1H), 6.96–6.92 (m, 1H); ^13^C-NMR (75 MHz, DMSO-*d*_6_): δ 167.0, 158.6 (d, *^1^J* = 239.7 Hz), 159.1, 150.2, 146.3, 142.3, 140.9, 133.8, 131.4, 129.1, 129.0, 127.9, 127.8, 122.7, 122.2, 120.8, 120.21 (d, *^2^J* = 24.6 Hz), 120.22 (d, *^3^J* = 9.7 Hz) 118.7, 118.54, 118.53, 116.8, 114.8, 114.5 (d, *^2^J* = 24.1 Hz). HRMS (ESI-FTMS) Mass (*m*/*z*): calcd for C_26_H_15_Cl_2_FN_4_O_4_ [M + H]^+^ = 537.0527, 539.0499, found 537.0512, 539.0482.

*4-[4-[(E)-[(3-chloro-2-fluoro-phenyl)hydrazono]methyl]-3-(6-fluoro-2-oxo-3,8a-dihydrochromen-3-yl)pyrazol-1-yl]benzoic acid* (**17**). Orange; (418 mg, 80%) ^1^H-NMR (300 MHz, DMSO-*d*_6_): 10.39 (s, 1H), 9.06 (s, 1H), 8.33 (s, 1H), 8.14–8.08 (m, 5H), 7.75–7.72 (m, 1H), 7.60–7.58 (m, 2H), 6.95 (t, *J* = 7.6 Hz, 1H), 6.81–6.70 (m, 2H); ^13^C-NMR (75 MHz, DMSO-*d*_6_): δ 167.0, 158.6 (d, *^1^J* = 239.7 Hz), 159.1, 150.2, 144.6 (d, *^1^J* = 240 Hz), 146.2, 142.3, 142.2, 135.2 (d, *^3^J* = 9.3 Hz), 133.1, 131.4, 129.1, 128.0, 125.4, 125.3, 123.0, 120.7, 120.3, 120.1 (d, *^3^J* = 8.5 Hz), 119.9 (d, *^3^J* = 7.3 Hz), 118.7 (d, *^3^J* = 13.7 Hz), 118.5, 114.5 (d, *^2^J* = 24.4 Hz), 112.5. HRMS (ESI-FTMS) Mass (*m*/*z*): calcd for C_26_H_15_ClF_2_N_4_O_4_ [M + H]^+^ = 521.0823, 523.0796, found 521.0806, 523.0783.

*4-[4-[(E)-[(3-chloro-4-fluoro-phenyl)hydrazono]methyl]-3-(6-fluoro-2-oxo-3,8a-dihydrochromen-3-yl)pyrazol-1-yl]benzoic acid* (**18**). Orange; (496 mg, 95%) ^1^H-NMR (300 MHz, DMSO-*d*_6_): 10.36 (s, 1H), 8.99 (s, 1H), 8.32 (s, 1H), 8.11–8.03 (m, 4H), 7.85 (s, 1H), 7.74–7.71 (m, 1H), 7.58–7.56 (m, 2H), 7.10 (t, *J* = 9.0 Hz, 1H), 6.83–6.80 (m, 1H), 6.70–6.65 (m, 1H); ^13^C-NMR (75 MHz, DMSO-*d*_6_): δ 167.1, 158.7 (d, *^1^J* = 239.3 Hz), 159.1, 150.8 (d, *^1^J* = 235.0 Hz), 150.2, 146.0, 143.0, 142.3, 142.1, 131.1, 130.5, 129.0, 127.9, 123.1, 120.8, 120.3 (d, *^3^J* = 8.6 Hz), 120.0 (d, *^2^J* = 24.4 Hz), 120.12, 118.8 (d, *^3^J* = 8.6 Hz), 118.4, 117.5 (d, *^2^J* = 21.9 Hz), 114.4 (d, *^2^J* = 24.1 Hz), 112.2, 111.9 (d, *^3^J* = 6.0 Hz). HRMS (ESI-FTMS) Mass (*m*/*z*): calcd for C_26_H_15_ClF_2_N_4_O_4_ [M + H]^+^ = 521.0823, 523.0796, found 521.0818, 523.0793.

*4-[3-(7-fluoro-2-oxo-3,8a-dihydrochromen-3-yl)-4-[(E)-[[4-(trifluoromethyl)phenyl]hydrazono]methyl]pyrazol-1-yl]benzoic acid* (**19**). Yellowish; (393 mg, 73%) ^1^H-NMR (300 MHz, DMSO-*d*_6_): 10.70 (s, 1H), 9.04 (s, 1H), 8.34 (s, 1H), 8.12–8.09 (m, 4H), 7.92 (s, 1H), 7.75–7.73 (m, 1H), 7.62–7.59 (m, 2H), 7.35–7.32 (m, 2H), 6.92–6.89 (m, 2H); ^13^C-NMR (75 MHz, DMSO-*d*_6_): δ 167.0, 158.6 (d, *^1^J* = 239.6 Hz), 159.2, 150.2, 148.5, 146.2, 142.3, 142.2, 131.8, 131.4, 129.2, 127.9, 126.6, 125.7 (q, *J* = 268.6 Hz) 122.9, 120.7, 120.2 (d, *^3^J* = 9.6 Hz), 118.6 (d, *^3^J* = 10.8 Hz), 118.3, 117.8, 114.5 (d, *^2^J* = 24.0 Hz), 111.7. HRMS (ESI-FTMS) Mass (*m*/*z*): calcd for C_27_H_16_F_4_N_4_O_4_ [M + H]^+^ = 537.1180, found 537.1177.

*4-[4-[(E)-[(4-cyanophenyl)hydrazono]methyl]-3-(7-fluoro-2-oxo-3,8a-dihydrochromen-3-yl)pyrazol-1-yl]benzoic acid* (**20**). Yellow; (436 mg, 88%) ^1^H-NMR (300 MHz, DMSO-*d*_6_): δ 10.86 (s, 1H), 9.08 (s, 1H), 8.36 (s, 1H), 8.13–8.06 (m, 4H), 7.95 (s, 1H), 7.76–7.73 (m, 1H), 7.65–7.55 (m, 2H), 7.46–7.43 (m, 2H), 6.90–6.87 (m, 2H); ^13^C-NMR (75 MHz, DMSO-*d*_6_): δ 167.1, 159.2, 158.6 (d, *^1^J* = 239.7 Hz), 150.2, 148.8, 146.2, 142.3, 133.8, 133.0, 131.4, 129.3, 128.1, 122.7, 120.54, 120.57, 120.2 (d, *^3^J* = 9.8 Hz), 120.0, 118.7, 118.59, 118.55, 114.5 (d, *^2^J* = 24.0 Hz), 112.1, 99.3. HRMS (ESI-FTMS) Mass (*m*/*z*): calcd for C_27_H_16_FN_5_O_4_ [M + H]^+^ = 494.1259, found 494.1252.

*4-[3-(7-fluoro-2-oxo-3,8a-dihydrochromen-3-yl)-4-[(E)-[(4-nitrophenyl)hydrazono]methyl]pyrazol-1-yl]benzoic acid* (**21**). Orange; (458 mg, 89%) ^1^H-NMR (30 0MHz, DMSO-*d*_6_): δ 11.2 (s, 1H), 9.08 (s, 1H), 8.35 (s, 1H), 8.11–8.02 (m, 5H), 7.94 (d, *J* = 9.2 Hz, 2H), 7.75–7.72 (m, 1H), 7.61–7.55 (m, 2H), 6.90–6.87 (m, 2H); ^13^C-NMR (75 MHz, DMSO-*d*_6_): δ 167.0, 158.6 (d, *^1^J* = 239.4 Hz), 159.2, 150.8, 150.3, 146.4, 142.3, 142.2, 138.4, 134.9, 131.4, 129.3, 128.4, 126.3, 122.5, 120.37, 120.32, 120.2, 120.1 (d, *^3^J* = 10.6 Hz), 118.3, 114.5 (d, *^2^J* = 24.4 Hz), 111.2. HRMS (ESI-FTMS) Mass (*m*/*z*): calcd for C_26_H_16_FN_5_O_6_ [M + H]^+^ = 514.1157, found 514.115.

*4-[4-[(E)-(dimethylhydrazono)methyl]-3-(6-fluoro-2-oxo-3,8a-dihydrochromen-3-yl)pyrazol-1-yl]benzoic acid* (**22**). Yellow; (333 mg, 79%) ^1^H-NMR (300 MHz, DMSO-*d*_6_): 8.75 (s, 1H), 8.27 (s, 1H), 8.12–8.06 (m, 4H), 7.73–7.70 (m, 1H), 7.54 (d, *J* = 6.4 Hz, 2H), 7.20 (s, 1H), 2.49 (s, 3H); ^13^C-NMR (75 MHz, DMSO-*d*_6_): δ 167.1, 158.5 (d, *^1^J* = 239.1 Hz), 159.1, 150.3, 146.1, 142.5, 141.8, 131.4, 128.8, 125.7, 125.5, 122.9, 122.3, 120.3 (d, *^3^J* = 9.6 Hz), 119.7 (d, *^2^J* = 24.8 Hz), 118.5 (d, *^3^J* = 8.7 Hz), 118.3, 114.3 (d, *^2^J* = 24.1 Hz), 42.9. HRMS (ESI-FTMS) Mass (*m*/*z*): calcd for C_22_H_17_FN_4_O_4_ [M + H]^+^ = 421.1307, found 421.1298.

*4-[3-(7-hydroxy-2-oxo-chromen-3-yl)-4-[(E)-(phenylhydrazono)methyl]pyrazol-1-yl]benzoic acid* (**23**). Brown; (370 mg, 79%) ^1^H-NMR (300 MHz, DMSO-*d*_6_): δ 10.17 (s, 1H), 8.94 (s, 1H), 8.22 (s, 1H), 8.08 (s, 5H), 7.81 (s, 1H), 7.67 (d, *J* = 8.4 Hz, 1H), 7.04 (t, *J* = 7.9 Hz, 2H), 6.83 (t, *J* = 14.7 Hz, 4H), 6.65 (t, *J* = 7.0 Hz, 1H); ^13^C-NMR (75 MHz, DMSO-*d*_6_) δ 167.1, 162.2, 159.9, 155.9, 146.9, 145.7, 143.7, 142.5, 131.4, 130.7, 129.5, 129.3, 128.8, 126.8, 121.2, 118.7, 118.3, 117.0, 114.0, 112.1, 111.8, 102.4. HRMS (ESI-FTMS) Mass (*m*/*z*): calcd for C26H18N4O5 [M + H]^+^ = 467.1350, found 467.1352.

*4-[4-[(E)-[(3-fluorophenyl)hydrazono]methyl]-3-(7-hydroxy-2-oxo-chromen-3-yl)pyrazol-1-yl]benzoic acid* (**24**). Brownish; (428 mg, 86%) ^1^H-NMR (300 MHz, DMSO-*d*_6_): δ 10.95 (br s, 1H), 10.54 (s, 1H), 9.00 (s, 1H), 8.22 (s, 1H), 8.08 (s, 5H), 7.87 (s, 1H), 7.66 (d, *J* = 8.1 Hz, 1H), 7.07–7.06 (m, 1H), 6.90–6.86 (m, 2H), 6.66–6.60 (m, 2H), 6.41 (t, *J* = 8.1 Hz, 1H); ^13^C-NMR (75 MHz, DMSO-*d*_6_) δ 167.0, 163.8 (d, *^1^J* = 238.9 Hz), 162.4, 159.9, 155.9, 147.7 (d, *^3^J* = 11.0 Hz), 147.0, 143.8, 142.5, 131.4, 130.8, 130.7, 130.6, 128.9, 127.2, 120.9, 118.3, 116.7, 114.1, 111.7, 108.2, 104.7 (d, *^2^J* = 21.7 Hz), 102.4, 98.5 (d, *^2^J* = 26.0 Hz). HRMS (ESI-FTMS) Mass (*m*/*z*): calcd for C_26_H_17_FN_4_O_5_ [M + H]^+^ = 485.1256, found 485.1257.

*4-[4-[(E)-[(4-chlorophenyl)hydrazono]methyl]-3-(7-hydroxy-2-oxo-chromen-3-yl)pyrazol-1-yl]benzoic acid* (**25**). Brown; (357 mg, 71%) ^1^H-NMR (300 MHz, DMSO-*d*_6_): δ 10.82 (br s, 1H), 10.32 (s, 1H). 8.95 (s, 1H), 8.22 (s, 1H), 8.06 (s, 4H), 7.80 (s, 1H), 7.66 (d, *J* = 8.2 Hz, 1H), 7.06 (d, *J* = 8.5 Hz, 3H), 6.83–6.81(m, 4H); ^13^C-NMR (75 MHz, DMSO-*d*_6_) δ 167.1, 162.2, 159.9, 156.0, 147.0, 144.6, 143.8, 142.5, 131.4, 130.7, 130.4, 129.0, 128.9, 127.0, 121.9, 120.9, 118.3, 116.7, 114.0, 113.5, 111.8, 102.4. HRMS (ESI-FTMS) Mass (*m*/*z*): calcd for C_26_H_17_ClN_4_O_5_ [M + H]^+^ = 501.0960, 503.0933, found 501.0962, 503.0932.

*4-[4-[(E)-[(4-bromophenyl)hydrazono]methyl]-3-(7-hydroxy-2-oxo-chromen-3-yl)pyrazol-1-yl]benzoic acid* (**26**). Brown; (437 mg, 80%) ^1^H-NMR (300 MHz, DMSO-*d*_6_): δ 10.88 (br s, 1H), 10.39 (s, 1H), 8.96 (s, 1H), 8.22 (s, 1H), 8.07 (s, 5H), 7.83 (s, 1H), 7.67 (d, *J* = 7.9 Hz, 1H), 7.18 (d, *J* = 7.8 Hz, 2H), 6.90–6.78 (m, 4H); ^13^C-NMR (75 MHz, DMSO-d_6_) δ 167.0, 162.3, 159.9, 155.9, 147.0, 145.0, 143.8, 142.5, 131.8, 131.4, 130.7, 130.4, 128.8, 127.0, 120.9, 118.3, 116.7, 114.1, 114.0, 111.8, 109.4, 102.4. HRMS (ESI-FTMS) Mass (*m*/*z*): calcd for C_26_H_17_BrN_4_O_5_ [M + H]^+^ = 545.0455, 547.0436, found 545.0456, 547.0435.

*4-[4-[(E)-[(2,5-difluorophenyl)hydrazono]methyl]-3-(7-hydroxy-2-oxo-chromen-3-yl)pyrazol-1-yl]benzoic acid* (**27**). Brownish; (459 mg, 91%) ^1^H-NMR (300 MHz, DMSO-*d*_6_): δ 10.94 (br s, 1H), 10.33 (s, 1H), 9.03 (s, 1H), 8.20 (s, 1H), 8.13–8.07 (m, 5H), 7.65 (d, *J* = 8.3 Hz, 1H), 7.12–7.04 (m, 1H), 6.90–6.84 (m, 3H), 6.44–6.39 (m, 1H); ^13^C-NMR (75 MHz, DMSO-*d*_6_) δ 167.0, 162.4, 159.7 (d, *^1^J* = 235.9 Hz), 159.8, 155.9, 145.5 (d, *^1^J* = 241.0 Hz), 147.0, 142.4, 135.2 (m), 133.5, 131.4, 130.6, 128.9, 127.7, 120.6, 118.4, 116.6, 116.4, 116.2 (d, *^3^J* = 9.6 Hz), 114.1, 111.6, 103.7 (d, *^2^J* = 25.0 Hz), 102.4, 100.4 (d, *^2^J* = 29.2 Hz). HRMS (ESI-FTMS) Mass (*m*/*z*): calcd for C_26_H_16_F_2_N_4_O_5_ [M + H]^+^ = 503.1162, found 503.1158.

*4-[4-[(E)-[(3,4-difluorophenyl)hydrazono]methyl]-3-(7-hydroxy-2-oxo-chromen-3-yl)pyrazol-1-yl]benzoic acid* (**28**). Black; (378 mg, 75%) ^1^H-NMR (300 MHz, DMSO-*d*_6_): δ 10.41 (s, 1H), 8.99 (s, 1H), 8.22 (s, 1H), 8.07 (s, 4H), 7.83 (s, 1H), 7.66 (d, *J* = 8.2 Hz, 1H), 7.17–7.07 (m, 1H), 6.88–6.76 (m, 3H), 6.57 (s, 1H); ^13^C-NMR (75 MHz, DMSO-*d*_6_) δ 167.1, 162.3, 159.9, 155.9, 152.1 (dd, *J* = 13.0, 254.0 Hz), 146.9, 144.4 (dd, *J* = 12.9, 246.3 Hz), 143.8, 142.4, 141.2 (d, *^3^J* = 12.8 Hz), 131.4, 130.7, 128.8, 127.3, 120.8, 119.3, 118.3, 118.0 (d, *^3^J* = 17.5 Hz), 116.7, 114.1, 111.7, 107.7, 102.4, 100.2 (d, *^2^J* = 21.9 Hz). HRMS (ESI-FTMS) Mass (*m*/*z*): calcd for C_26_H_16_F_2_N_4_O_5_ [M + H]^+^ = 503.1162, found 503.1162.

*4-[4-[(E)-[(2,4-dichlorophenyl)hydrazono]methyl]-3-(7-hydroxy-2-oxo-chromen-3-yl)pyrazol-1-yl]benzoic acid* (**29**). Brownish; (483 mg, 90%) ^1^H-NMR (300 MHz, DMSO-*d*_6_): δ 10.94 (br s, 1H), 9.96 (s, 1H), 9.04 (s, 1H), 8.24 (d, *J* = 5.7 Hz, 2H), 8.09 (s, 5H), 7.67 (d, *J* = 8.1 Hz, 1H), 7.40 (s, 1H), 7.23 (d, *J* = 8.7, 1H), 7.00 (d, *J* = 8.9 Hz, 1H) 6.92–6.88 (m, 2H); ^13^C-NMR (75 MHz, DMSO-*d*_6_) δ 167.0, 162.4, 159.9, 155.9, 147.3, 144.0, 142.4, 141.0, 134.2, 131.4, 130.7, 128.9, 127.9, 127.3, 122.1, 120.7, 118.4, 116.7, 116.4, 115.0, 114.1, 111.7, 102.4. HRMS (ESI-FTMS) Mass (*m*/*z*): calcd for C_26_H_16_Cl_2_N_4_O_5_ [M + H]^+^ = 535.0571, 537.0543, found 535.0570, 537.0545.

*4-[4-[(E)-[(3-chloro-2-fluoro-phenyl)hydrazono]methyl]-3-(7-hydroxy-2-oxo-chromen-3-yl)pyrazol-1-yl]benzoic acid* (**30**). Brown; (437 mg, 84%) ^1^H-NMR (300 MHz, DMSO-*d*_6_): δ 10.33 (s, 1H), 9.00 (s, 1H), 8.32 (s, 1H), 8.11–8.07 (m, 5H), 7.66 (d, *J* = 8.1 Hz, 1H), 7.08 (s, 1H), 6.89–6.79 (m, 4H); ^13^C-NMR (75 MHz, DMSO-*d*_6_) δ 167.1, 162.3, 159.9, 155.9, 147.1, 144.7 (d, *^1^J* = 240.2 Hz), 143.8, 142.4, 135.3 (d, *^3^J* = 9.3 Hz), 133.5, 131.4, 130.7, 128.9, 127.5, 125.4, 120.6, 119.9 (d, *^3^J* = 14.2 Hz), 118.7, 118.4, 116.7, 114.1, 112.6, 111.7, 102.4. HRMS (ESI-FTMS) Mass (*m*/*z*): calcd for C_26_H_16_ClFN_4_O_5_ [M + H]^+^ = 519.0866, 521.0839, found 519.0869, 521.0844.

*4-[4-[(E)-[(3-chloro-4-fluoro-phenyl)hydrazono]methyl]-3-(7-hydroxy-2-oxo-chromen-3-yl)pyrazol-1-yl]benzoic acid* (**31**). Brown; (380 mg, 73%) ^1^H-NMR (300 MHz, DMSO-*d*_6_): δ 10.42 (s, 1H), 8.99 (s, 1H), 8.22 (s, 1H), 8.10–8.07 (m, 5H), 7.84 (s, 1H), 7.66 (d, *J* = 7.8 Hz, 1H), 7.12 (t, *J* = 9.5 Hz, 1H), 6.94–6.85 (m, 3H), 6.75 (br s, 1H); ^13^C-NMR (75 MHz, DMSO-*d*_6_) δ 167.0, 162.3, 159.9, 155.9, 150.8 (d, *^1^J* = 234.6 Hz), 146.9, 143.7, 143.2, 142.5, 131.4, 130.7 (d, *^3^J* = 14.9 Hz), 128.9, 127.4, 120.7, 120.2 (d, *^2^J* = 18.1 Hz), 119.4, 118.3, 117.4 (d, *^2^J* = 21.6 Hz), 116.8, 114.0, 112.4, 111.9 (d, *^3^J* = 6.8 Hz), 111.7, 102.6. HRMS (ESI-FTMS) Mass (*m*/*z*): calcd for C_26_H_16_ClFN_4_O_5_ [M + H]^+^ = 519.0866, 521.0839, found 519.0866, 521.0837.

*4-[3-(7-hydroxy-2-oxo-3,8a-dihydrochromen-3-yl)-4-[(E)-[[4-(trifluoromethyl)phenyl]hydrazono]methyl]pyrazol-1-yl]benzoic acid* (**32**). Yellow; (445 mg, 83%) ^1^H-NMR (300 MHz, DMSO-*d*_6_): δ 10.7 (s, 1H), 9.00 (s, 1H), 8.24 (s, 1H), 8.08 (s, 4H), 7.90 (s, 1H), 7.68 (d, *J* = 8.3 Hz, 1H), 7.36 (d, *J* = 8.5 Hz, 1H), 6.96–6.86 (m, 3H); ^13^C-NMR (75 MHz, DMSO-*d*_6_) δ 167.1, 162.3, 159.9, 156.0, 148.6, 147.1, 143.9, 142.4, 132.1, 131.4, 130.8, 129.0, 127.4, 127.2, 126.6, 123.6, 120.6, 118.6-118.1 (m), 116.6, 114.1, 111.8, 111.7, 102.3. HRMS (ESI-FTMS) Mass (*m*/*z*): calcd for C_27_H_17_F_3_N_4_O_5_ [M + H]^+^ = 535.1224, found 535.1223.

*4-[4-[(E)-[(4-cyanophenyl)hydrazono]methyl]-3-(7-hydroxy-2-oxo-3,8a-dihydrochromen-3-yl)pyrazol-1-yl]benzoic acid* (**33**). Yellow; (399 mg, 81%) ^1^H-NMR (300 MHz, DMSO-*d*_6_): δ 10.83 (s, 1H), 9.02 (s, 1H), 8.25 (s,1H), 8.08 (s, 4H), 7.92 (s, 1H), 7.67 (d, *J* = 8.3 Hz, 1H), 7.46 (d, *J* = 8.7 Hz, 2H), 6.95–6.85 (m, 4H); ^13^C-NMR (75 MHz, DMSO-*d*_6_) δ 167.1, 162.3, 159.9, 156.0, 148.9, 147.2, 143.9, 142.4, 133.8, 133.3, 131.4, 130.8, 129.0, 127.6, 120.6, 120.4, 118.4, 116.4, 114.1, 112.1, 111.8, 102.4, 99.2. HRMS (ESI-FTMS) Mass (m/z): calcd for C_27_H_17_N_5_O_5_ [M + H]^+^ = 492.1302, found 492.1300.

*4-[3-(7-hydroxy-2-oxo-3,8a-dihydrochromen-3-yl)-4-[(E)-(methoxycarbonylhydrazono)methyl]pyrazol-1-yl]benzoic acid* (**34**). Brownish; (324 mg, 72%) ^1^H-NMR (300MHz, DMSO-*d*_6_): δ 10.97 (br s, 1H), 8.95 (s, 1H), 8.24 (s, 2H), 8.06 (s, 4H), 7.95 (s, 1H), 7.66 (d, *J* = 8.4 Hz, 1H), 6.87–6.81 (m, 2H); ^13^C-NMR (75 MHz, DMSO-*d*_6_) δ 167.4, 162.6, 160.0, 156.0, 154.2, 147.7, 144.3, 142.0, 131.3, 130.8, 130.3, 127.5, 119.6, 118.9, 118.6, 115.6, 114.1, 111.6, 102.4, 52.2. HRMS (ESI-FTMS) Mass (*m*/*z*): calcd for C_22_H_16_N_4_O_7_ [M + H]^+^ = 449.1092, found 449.1107.

### 3.2. Culturing of Bacteria

Bacterial cultures were maintained on tryptic soy agar (TSA) slants. Bacteria prepared for MIC testing or plated for time kill assays were grown on blood agar (TSA with 5% sheep blood) plates. Bacteria grown in liquid culture including 96-well plates for MIC testing were grown in Mueller Hinton Broth (cation adjusted, CAMHB). Bacterial dilutions were made in normal saline or phosphate buffered saline (PBS)

### 3.3. Minimum Inhibitory Concentration (MIC)

The broth microdilution method was utilized to determine MIC values of coumarin-substituted pyrazole derivatives against different clinically important *S. aureus* strains, *B. subtilis* and *S. epidermidis* according to the guidelines outlined by the Clinical and Laboratory Standards Institute (CLSI) as reported in our recent papers [9]. The starting concentration of compounds for MIC determination was 50 μg/mL serially diluted down the wells and the MIC values were recorded in duplicates in three independent experiments on different days.

### 3.4. Time Kill Assay

Time kill assay was performed against two strains of bacteria (*A. baumannii* ATCC 19606 and *S. aureus* ATCC 33599) using our most potent compounds. Bacteria in the logarithmic growth phase (OD600 ~2.00) were diluted to 1.5 × 10^6^ colony-forming units (CFU/mL) and exposed to concentrations equivalent to 4×MIC (in triplicates) of tested compounds, vancomycin and colistin as control drugs for *S. aureus* and *A. baumannii* respectively. Aliquots were collected from treatment after 0, 2, 4, 6, 8, 10, 12, and 24 h of incubation at 35 °C and subsequently serially diluted in PBS. The diluted aliquots were then transferred to blood agar plates and incubated at 35 °C for 18–20 h before viable CFU/mL was determined by using the 6 × 6 drop plate method.

### 3.5. Biofilm Inhibition Assay

The biofilm-forming strain *S. aureus* ATTC 25923 was grown in the presence of four different coumarin-substituted pyrazole derivatives with the lowest MIC values to determine if the compounds would interfere with biofilm formation. An overnight culture of bacteria was suspended in PBS solution to match the 0.5 McFarland standard and then diluted 1:100 into CAMHB with 1% glucose to give an approximate final concentration of 1 × 10^6^ CFU/mL. Bacterial broth suspension (195 μL) was transferred to each well in 96-well polystyrene flat bottom plate. Compounds (5 μL) at 2 × MIC, MIC, and 0.5 × MIC were added to wells in triplicate along with broth only, and bacteria along with DMSO controls, and plates were incubated at 35 °C for 24 h. After incubation, the contents of the wells were removed and wells were washed with 1×PBS solution three times to remove any planktonic cells. The plate was dried in an oven at 60 °C for about 15 min and 0.1% (*w*/*v*) crystal violet (250 μL) was added to each well and left for 15 min for staining biofilms. Excess crystal violet was removed by draining and washing three times with deionized water, and the plate was again dried in oven for 10 min. After drying, 33% acetic acid (250 μL) was added to each well to dissolve the stained biofilm. The optical density of the solubilized crystal violet in each well was measured at 620 nm using a Bio Tek^TM^ Cytation^TM^ 5 plate reader.

### 3.6. Biofilm Destruction Assay

This assay was performed to test whether our four most effective compounds could destroy the preformed biofilm in vitro. Bacterial cultures were prepared as for the biofilm inhibition test (195 µL culture per well) but without addition of compounds. Bacteria and growth medium only controls were incubated overnight at 35 °C for 24 h to allow bacterial formation of enough biofilm. After incubation, soluble and suspended well contents were carefully removed, and wells were washed with sterile PBS solution to remove any unadhered cells. Next, 195 μL sterile CAMHB with 1% glucose was added to each well with 5 μL of 2 × MIC, MIC, and 0.5 × MIC concentrations of compounds or DMSO in triplicate and the plate was incubated at 35 °C for 24 h. This extra 24 h incubation was required in order to provide enough time for compounds to destroy the preformed biofilm in the plate. After incubation, washing, drying, staining, dissolving stained dye, and measuring optical density in a plate reader were performed as described above.

### 3.7. Processing of Data

As these biofilm assays were performed in triplicates mean and standard deviation of plate reading data were processed. Results were expressed as a percentage by using the formula:

Percentage biofilm inhibition/destruction $=\left[1-\frac{ODcompound-ODbroth}{ODdmso-ODbroth}\right]$ where, OD_compound_ = optical density of well with compound, OD_broth_ = OD of well with broth only, OD_dmso_ = OD of well with bacteria broth + DMSO.

The data were processed and represented in graphical form in Microsoft^®^ Excel^®^ for Office 365 MSO.

## 4. Conclusions

Because of the ease of synthesis without column purification or work-up, we have reported the synthesis of new hydrazone derivatives of coumarin-derived pyrazoles. We synthesized 31 new pyrazole derivatives. These new molecules were tested against several bacterial strains, and we found several molecules, which showed promising results with MIC values as low as 1.56 µg/mL. We found that fluoro-substituted compounds are more potent than the hydroxy-substituted compounds. Potent molecules are growth inhibitor of MRSA biofilms and eliminated the preformed biofilm more efficiently than the positive control, vancomycin. One of the potent molecules (**19**) showed very mild toxicity when comparing the IC_50_ against HEK293 cells to the MIC against bacteria. Future direction will be the synthesis of molecules related to the potent compounds to target bacterial biofilms.

## 5. Patents

Alam, M. A. Antimicrobial agents and the method of synthesizing the antimicrobial agents. US Patent. 10,596,153, 2020.

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
