# Peer review of "Synthesis and Antimicrobial Studies of Coumarin-Substituted Pyrazole Derivatives as Potent Anti-Staphylococcus aureus Agents"

_molecules, 2020, doi:10.3390/molecules25122758_

Round 1

Reviewer 1 Report

The paper describes the synthesis and characterization of 31 Coumarin-Pyrazole derivatives as potential anti-Staphylococcus aureus agents. The scheme of synthesis has already been published.

In this paper the authors propose additional new derivatives. All the products are well characterized and evaluated in terms of their antimicrobial activities.

Although the topic is of a high level of interest, in my opinion in its present form there is too little novelty for publication in Molecules. The authors should concentrate their efforts on proposing a structure activity relationship. Based on their previous studies and on literature data, they could improve the quality of paper by investigating more deeply the short sequence suggested in the conclusion.“We found that the fluoro-substituted compounds are more potent than the hydroxy-substituted

 compounds.”

Author Response

We thank the reviewer the valuable comments. 

Reviewer 2 Report

The manuscript deals with the synthesis and antimicrobial studies of coumarin-substituted pyrazole derivatives. The synthetic procedure for achieving the compounds was not followed by column purification, which is helpful for the methodology. In addition, the yields of the products were interesting.

Regarding the antibacterial assays, the activity of the N,N-diphenyl substituted compound was highlighted. This molecule inhibited the growth of all the tested Gram-positive bacterial strains, and also some methicillin-resistant S. aureus strains. Interestingly, replacing one phenyl group with a benzyl group eliminated the antibacterial activity of the resultant molecule. At this point, the authors should have provided a deeper discussion about this topic through comparison with literature data.
For instance, the authors could have compared the antibacterial activities of other compounds containing N,N-diphenyl and N,N-dibenzyl moieties.

At the Material and Methods section: the authors have said that the Minimum Inhibitory Concentration assay was carried out according to their previous papers (page 13). I suggest informing the assayed concentrations of the compounds and the number of replicates.

Author Response

We thank the reviewer for valuable comments.

Round 2

Reviewer 1 Report

The authors addressed in full all my concerns and suggestions. From my side, I can now recommend the acceptance of the paper in Molecules.

Author Response

We appreciate your feedback and valuable time to improve the manuscript.